# Evaluation of Two Active System Encapsulant Matrices with Quercetin and *Bacillus clausii* for Functional Foods

**DOI:** 10.3390/polym14235225

**Published:** 2022-12-01

**Authors:** Hector Alfonso Enciso-Huerta, Miguel Angel Ruiz-Cabrera, Laura Araceli Lopez-Martinez, Raul Gonzalez-Garcia, Fidel Martinez-Gutierrez, Maria Zenaida Saavedra-Leos

**Affiliations:** 1Facultad de Ciencias Químicas, Universidad Autónoma de San Luis Potosí, Av. Dr. Manuel Nava 6, San Luis Potosí 78210, Mexico; 2Coordinación Académica Región Altiplano Oeste, Universidad Autónoma de San Luis Potosí, Salinas de Hidalgo 78600, Mexico; 3Coordinación Académica Región Altiplano, Universidad Autónoma de San Luis Potosí, 11 Carretera Cedral Km, 5+600 Ejido San José de las Trojes, Matehuala 78700, Mexico

**Keywords:** functional, food, inulin, lactose, *Bacillus clausii*

## Abstract

Currently, demand for functional foods is increasing in the public interest in order to improve life expectations and general health. Food matrices containing probiotic microorganisms and active compounds encapsulated into carrier agents are essential in this context. Encapsulation via the lyophilisation method is widely used because oxidation reactions that affect physicochemical and nutritional food properties are usually avoided. Encapsulated functional ingredients, such as quercetin and *Bacillus clausii*, using two carrier agents’ matrices—I [inulin (IN), lactose (L) and maltodextrin (MX)] and II [arabic (A), guar (G), and xanthan (X) gums)]—are presented in this work. A D-optimal procedure involving 59 experiments was designed to evaluate each matrix’s yield, viability, and antioxidant activity (AA). Matrix I (33.3 IN:33.3 L:33.3 MX) and matrix II (33.3 A:33.3 G:33.3 X) exhibited the best yield; viability of 9.7 log10 CFU/g and 9.73 log10 CFU/g was found in matrix I (using a ratio of 33.3 IN:33.3 L:33.3 MX) and matrix II (50 G:50 X), respectively. Results for the antioxidant capacity of matrix I (100 IN:0 L:0M X) and matrix II (0 A:50 G:50 X) were 58.75 and 55.54 (DPPH* scavenging activity (10 µg/mL)), respectively. Synergy between matrices I and II with use of 100IN:0L:OMX and 0A:50G:50X resulted in 55.4 log10 CFU/g viability values; the antioxidant capacity was 9. 52 (DPPH* scavenging activity (10 µg/mL). The present work proposes use of a carrier agent mixture to produce a functional ingredient with antioxidant and probiotic properties that exceed the minimum viability, 6.0 log10 CFU/g, recommended by the FAO/WHO (2002) to be probiotic, and that contributes to the recommended daily quercetin intake of 10–16 mg/day or inulin intake of 10–20 g/day and dietary fibre intake of 25–38 g per day.

## 1. Introduction

A functional food (FF) is defined by the European Society for Clinical Nutrition and Metabolism guide as an enriched food with ingredients, nutrients, or additional compounds intended to manifest specific benefits to health. In the last decade, FF production has become an important biotechnology industry, given growing consumer interest in improving life expectancy and healthy due to raising of awareness about prevention of certain diseases such as diabetes, cancer, and Alzheimer’s [1,2]. Antioxidants are commonly compounds added to other nutritional particles to promote synergism: e.g., vitamin C to regenerate the vitamin E tocopheryl radical after its oxidation [3]. Additionally, antioxidants are added to suppress lipid oxidation, increase products’ shelf life, and reduce free radical concentrations generated in organisms [4]. Flavonoids such as quercetin are antioxidants found in apples, grapes, beans, broccoli, red onion, tomatoes, oilseeds, flowers, tea leaves, and Ginkgo biloba. However, though recommended ingestion of quercetin is 1 g per day, the average consumption is about 10–16 mg [5]. Different molecular mechanisms have been reported in treatment of various diseases. For example, in allergic asthma, the compound showed inhibition of *MUC5AC* gene expression in NCI-H292 cells, triggering human nasal mucosa anti-secretory agents that prevented mucosa secretion in epithelial cells while maintaining a normal ciliary movement [6]. Ingestion of 150–730 mg of quercetin per day over four weeks resulted in antihypertensive action, reducing systolic and diastolic pressure in patients in the first stage of hypertension [7]. Similarly, patients with metabolic syndrome who consumed a daily dose of 150 mg of quercetin over five weeks significantly reduced their systolic pressure. An in vitro study performed by Reyes-Farias and Carrasco-Pozo [8] showed that quercetin acts as an antiviral agent against HIV, inhibiting integrase, protease, and inverse transcriptase enzymes.

Other important compounds found in FFs are probiotics, which confer benefits to health through production of biliary enzymes, organic acid, satiety hormones, and immune system modulation. These microorganisms also improve antibody response, improve substrate competence against pathogenic organisms, and interact with microbiota [9]. An important probiotic employed in food enrichment is *Bacillus clausii* (*B. clausii*): an anaerobic gram-positive bacterium capable of generating spores and intestinal colonisers [10]. Moreover, *B. clausii* is resistant to heat, gastric pH conditions, and antibiotics. Nevertheless, its optimal growth conditions are 40 °C and a pH of 9.0. De Castro et al. [11] treated acute infant diarrhoea (by viral cause or associated with antibiotics) with *B. clausii*, showing that its consumption over seven days reduced disease duration, gastrointestinal symptoms, and evacuation frequency. Plomer et al. [12] used *B. clausii* to reduce adverse effects of treatment of *Helicobacter pylori*: a pathology, usually treated with antibiotics, that causes nausea, inflammation, vomit, and diarrhoea, triggering treatment failure and bacterial resistance.

A microencapsulation process is employed to conserve active ingredient properties susceptible to suffering damage under processing or environmental conditions. Major environmental conditions that could affect ingredient activity include atmospheric oxygen, pH, humidity, light irradiation, and high temperature exposure. The microencapsulation technique involves use of an encapsulating material that maintains its microstructural integrity in aggressive environments in which active ingredients may lose their functions. Nutraceutical and functional ingredients such as antioxidants, vitamins, minerals, lipids, and probiotics have been microencapsulated via different methodologies [13,14,15]. Polysaccharides, lipids, and proteins are examples of different compounds employed as wall materials in microencapsulation [16]. Inulin (IN) is a non-digestible polymer, presenting fructose linear chains with a terminal group consisting of glucose molecules joined through β-(2,1) bonds. This molecule is found in many vegetables, fruits, and cereals; is structurally considered a short-chain carbohydrate with a polymerisation rate between 1 and 60 repetitive units per molecule; and is a water-soluble biopolymer [17]. On the other hand, maltodextrin (MX) is a polysaccharide derived from starch acidic hydrolysis, presenting a nutritional contribution of only 4 calories per gram. This molecule is commercialised in a wide range of molecular weight distributions (MWDs), each with different thermal properties and potential applications. Recently, Saavedra-Leos et al. [18] reported use of a set of four MXs as carrying agents in spray-drying of blueberry juice–maltodextrin (BJ–MX); they set application limits of the maltodextrins based on the MWDs. Lactose in its mono-hydrated form has been widely used as an excipient in order to facilitate administration of drugs, especially those that target the lungs [19]. Gum arabic (GA) is a naturally occurring polysaccharide, obtained from resin of certain varieties of Acacia (*Mimosoidae* subfamily), of low viscosity, solubility, and emulsion formation; it can act as an encapsulating agent in combination with other agents, such as xanthan gum, MX, or modified starch. GA is a carbohydrate extracted from the plant *Cyamopsis tetragonoloba*, whose main characteristics are to hydrate rapidly in cold water and produce highly viscous solutions [20]. It is used as a thickener and a viscosity modifier in a wide variety of processed foods, such as ice cream, cheese, bread, meats, dressings, and sauces, or pharmaceuticals and cosmetics [21]. Xanthan gum (XG) is a natural polysaccharide produced through fermentation of *Xanthomonas campestris*. It is highly soluble in water, producing high viscosity; stable in alkaline or acidic conditions; and widely used as a stabiliser in foods such as creams, artificial juices, sauces, syrups, ice cream toppings, meat, poultry and fish. An investigation performed by Lombardo and Villares [22] demonstrated that cellulose, MX, IN, and starch polysaccharides have been used as carrier agents to improve rigidity of microencapsulations. Other investigations [14,15] performed freeze-drying encapsulation of ethanolic *Elsholtzia ciliata* and *Lactobacillus plantarum* extracts using different combinations of gum A, MX, lactose, and skimmed milk; these investigations obtained yields in the range of 90–100%.

Production of foods that contain probiotics usually employs spray-drying and lyophilisation techniques. The latter is considered by the food, pharmaceutical, and biotechnological industries as a drying process capable of stabilizing and preserving products through reduction of loss of unstable compounds. Consequently, it is the preferred methodology of preserving aromas, flavours, and nutritional compounds [23]. In contrast to the spray-drying process, because of the low temperature at which lyophilisation is carried out, oxidation reactions are not catalysed, thus preventing physicochemical and nutritional food damage [24]. The freeze-drying process consists of freezing the product at −40 °C and sublimating the ice at sub-atmospheric pressures [25]. A study performed by Gümüşay et al. [26] compared different drying methods, with the aim to obtain higher content of phenolic compounds, ascorbic acid and antioxidant activity from tomatoes. These authors observed that freeze-drying resulted in about double phenolic compounds compared to those yielded by other drying methods. Rockinger et al. [27] reviewed current approaches to cell preservation through freeze-drying and found that stability of cells is achieved by cryopreservation at sub-zero temperatures (−130 °C). Solid-state water is removed through sublimation, and no residual moisture remaining in the solid is enough to allow molecular movement and biochemical reactions; this, in turn, may preserve the food product and promote longer storage periods.

Taking into consideration the importance of antioxidants and probiotics in health, the objective of this work is to evaluate a combination of three carrier agents (inulin, lactose and maltodextrin) and three gums (xanthan, arabic and guar) in co-encapsulation of *Bacillus clausii* and quercetin in a functional food prepared via freeze-drying. A special cubic design of experiments employing the Scheffe mix model was implemented to achieve this purpose and to compare the extent of the response variables (viability of *B. clausii* and antioxidant activity of quercetin).

## 2. Materials and Methods

### 2.1. Materials

Commercial maltodextrin (MX) extracted from maize starch was acquired from INGREDION Mexico (Guadalajara, Mexico). The dextrose (DE) equivalent of MX was 10, with a molecular weight of 1625 g/mol and a polymerisation grade (DP) of 2–16 glucose units. Inulin (IN) was purchased from INGREDION Mexico (Guadalajara, Mexico). α-lactose monohydrate (L) (Lα·H2O, purity ≥ 99.9%) was purchased from Sigma-Aldrich Chemical Co (Toluca, Mexico).; methanol (MeOH, purity ≥ 99.8) was obtained from J.T. Baker (Guadalajara, Mexico). Gums arabic (A), guar (G) and xanthan (X) were obtained from INGREDION Mexico (Guadalajara, Mexico). The *Bacillus* strain (*B*. *clausii*) in sinuberase solution was purchased from Sanofi-Aventis Mexico, S.A. de C.V. (Coyocan, Mexico City, Mexico). Quercetin 3-D-Galactose (purity ≥ 99%) was acquired from Química Farmacéutica Esteroidal S.A de C.V. (Tlahuac, Mexico City, Mexico). Trypticase Soy Agar was obtained from Dickinson de México S.A. de C.V. (Mexico City, Mexico). Finally, analytical-grade 2,2-diphenyl-1-picrilhidrazile (DPPH) was obtained from Sigma–Aldrich Chemical Co (Toluca, Mexico).

### 2.2. Lyophilisation Preparation

In Table 1—the experimental design of two matrices for a special cubic x special cubic model—the resulting 59 tests performed in the laboratory were carried out in a random order. For each test, 100 g samples (*w*/*w*) were prepared.

Each mass fraction for matrices I and II was set according to the experimental design. The compounds of matrix I (10 g) and matrix II (1 g) were passed through a 1 mm sieve. Subsequently, 87 g deionised water was added and magnetically stirred at 35 °C for 5 min. Next, 1 g quercetin and 1 g *B. clausii* were added. The samples were stored in the dark at −80 °C. The microencapsulation process was carried out by sublimation in a freeze dryer (Ilshin Bio Base^®^ Model TFD8501, Gyeonggi-do, South Korea) under a vacuum pressure of 5 mTorr −65 °C for approximately 120 h. Yield was determined using Equation (1):(1)$$Yield=\left[\frac{SL}{SI}\right]*100\;$$

*SL* = solids recovered at the end of freeze-drying

*SI* = Initial solids (10 g matrix I + 1 g matrix II + 1 g quercetin + 1 g *B. clausii*)

### 2.3. Determination of Microbial Viability

Viability of *B. clausii* before and after the encapsulation process was determined through resuspension of 1 g of the microparticles obtained in 9 mL of saline solution (NaCl, 0.9% *w*/*v*). To break microcapsules, the suspension was agitated for 10 min with a vortex and incubated in a water bath for 10 min at 50 °C. Viable cells were analysed according to the method described by Miles et al. [28]. Briefly, dilutions of 1 × 10^−3^ a 1 × 10^−9^ performed in saline solution were sown on trypticase soy agar and incubated at 35 °C for 24 h. The evaluation was performed in triplicate and reported in colony-forming units per gram (CFU/g), using Equation (2):(2)$$Viability=\left[\frac{Number\;of\;colonies\;in\;box*dilution\;factor\;}{\;mL\;of\;sample\;sown\;}\right]\;$$

### 2.4. Antioxidant Activity (AA)

Quercetin antioxidant capacity was determined according to the method described by Brand-Williams et al. [29]. Briefly, 1.7 mL of alcoholic solution of *DPPH* (0.1 mmol DPPH/L) was mixed with 1.7 mL of microencapsulated suspension in which concentration of microencapsulation varied from 2.5 to 5 or 15 µg/mL. The mixture was left to stand in darkness for 30 min, and absorbance at 537 nm was measured using a spectrophotometer UV-Vis Evolution 220 (Thermo Scientific, Walthman, MA. USA). The sweep percentage was calculated using equation 3:(3)$$AA\;\left(\%DPPH\right)=\frac{A0-A30}{A0}\times 100\;$$
where *A*0 represents absorbance of blank solution (*DPPH* mixture and ethanol without microencapsulates) and *A*30 represents absorbance of *DPPH* solution and ethanol with microencapsulates after 30 min. Sweep activity was determined in triplicate for each sample.

### 2.5. Design of Experiments and Statistical Analysis

Two independent mixtures were tested. Matrix I consisted of inulin (IN), lactose (L), and maltodextrin (MX), while matrix II consisted of gums arabic (A), guar (G), and xanthan (X). The lower and upper levels of these variables were between 0 and 100 (wt %), and the sum of the components in each mixture was 100% for each trial. The response variables were yield (%), *Bc* (Log10 CFU/g), and antioxidant activity (DPPH at concentrations of 5, 10, and 30 (μg/g)). In this manner, a combined experimental design of two matrices for a special cubic x special cubic model was selected to evaluate the effect of each factor for each response variable. Table 1 shows the resulting 59 trials performed at the laboratory in a random order.

An analysis of variance (ANOVA) was performed for each response (yield, *Bc*, and antioxidant activity) at the significance level of 0.05, using Design-Expert^®^ Version 12 Software (trial version). The analysed Scheffe model (special cubic x special cubic) was written as Equation (4):Y = (α_1_A + α_2_B + α_3_C + α_4_AB + α_5_AC + α_6_BC + α_7_ABC) × (κ_1_D + κ_2_E + κ_3_F + κ_4_DE + κ_5_DF + κ_6_EF + κ_7_DEF) (4)
which is an expanded method that results in 49 adjustable parameters. In Table 2: ANOVA statistical analyse for each response observed

## 3. Results and Discussion

The ANOVA for capacity antioxidant and *B. clausii* response variables is discussed herein.

**Table 2 polymers-14-05225-t002:** Statistical ANOVA details for each response analysed.

Response	SST	SSR	SSE	DFT	DFR	DFE	F	P(F)	R^2
Antioxidant Capacity for 2,2 Difenil-1-Picrilhidrazil (DPPH) 5 µg/mL	1177.97	1053.83	124.14	58	48	10	1.77	0.1665	0.8946
Antioxidant Capacity for 2,2 Difenil-1-Picrilhidrazil (DPPH) 10 µg/mL	1218.56	1117.41	101.15	58	48	10	2.3	0.0778	0.9170
Antioxidant Capacity for 2,2 Difenil-1-Picrilhidrazil (DPPH) 30 µg/mL	2914.83	2738.77	176.06	58	48	10	3.24	0.0246	0.9396
*B. clausii*	1.38	1.3	0.076	58	48	10	3.55	0.0177	0.9445
SST:	Sum of Squares Total								
SSR:	Sum of Squares Regression								
SSE:	Sum of Squares Error								
DFT:	Degrees of Freedom Total								
DFR:	Degrees of Freedom of Regression								
DFE:	Degrees of Freedom of Error								
F:	Fisher’s Statistic								

### 3.1. Microencapsulation Performance

The microencapsulation technique comprised coating small particles to form capsules with unique properties and different morphologies, each of which could reach diameters from nanometres to millimetres, protect bioactive ingredients from adverse reactions, and improve functionality and bioavailability [30]. Lyophilisation is a microencapsulation process in which a previously frozen product (−40 °C) is lyophilised to ice sublimation at sub-atmospheric pressures. This work evaluated two different encapsulating matrices for two active systems (antioxidant and microorganism) in order to produce functional lyophilised food. Matrix I, containing IN, L, and MX, as shown in Figure 1a, featured an efficiency of 93.7% when IN was present at 66.71% in example experiment 54 (66.7 IN:16.7 L:16.7 MX); it featured an efficiency of 87.6% for IN at 100% in the case of experiment 49 (100 IN:0 L:0 MX). Matrix II, containing arabic (A), guar (G), and xanthan (X) gums, showed better efficiency, corresponding to 88.08% when A, G, and X gums were present in the same proportion: e.g., experiment 32 (33.3 A:33.3 G:33.3 X), as shown in Figure 1b. Our results are consistent with the report of Enache et al. [31], who employed the freeze-drying method of co-microencapsulation of black-currant-extract anthocyanins and lactic-acid bacteria, using inulin and chitosan as carrier agents. They reported a recovery efficiency of 95.46% ± 1.30% for inulin and 87.38% ± 0.48% for chitosan. Pudziuvelyte et al. [14] reported microencapsulate lyophilisation yield of the ethanolic extract of *Elsholtzia ciliata*, with six carrier agents at 20% concentration and mixes at 10% concentration; they employed arabic gum (GUM_E), maltodextrin (MALTO_E), resistant maltodextrin (RES_E), skimmed milk (SKIM_E), sodium caseinate (SOD_CAS_E), and beta-cyclodextrin (BETA_CYCL_E). These authors indicated that a higher yield was observable when they employed SKIM_E and MALTO_E, which showed 100% and 95% efficiency, respectively, for mixtures of an observed 100% yield in two situations: use of SKIM_E with MALTO_E and of GUM_E with BETA_CYCL_E. Sharifi et al. [15] performed analysis of co-microencapsulate Lactobacillus plantarum and phytosterol mixtures formed with β-sitosterol (49.54%), campesterol (26.12%), stigmasterol (19.1%), and brassicasterol (1.48%). Researchers used gum arabic (GA) (2.25% w/v) and whey protein isolate (WPI) (5% *w/v*) as encapsulating agents. They formed coacervates followed by two dry processing techniques—aspersion drying and lyophilisation—obtaining 58.62 ± 2.01% and 65.23 ± 0.51% yields, respectively. These results demonstrated improved performance with use of a dry freeze-drying process. The yield results contributed to a range of responses; for example, higher yields were obtained when the carrier agent used in matrix I had an IN value closer to 100 or when A:G:X demonstrated the same ratio for matrix II, and lower yields were obtained with use of only MX at 100 for matrix I, or, in the case of matrix II, with use of G 100.

### 3.2. Viability of B. clausii Microencapsulated

Viability of *B. clausii* was measured with the colony forming unit method (CFU/g). The results shown in Figure 2a were determined concerning the viability in matrix I. Experiments 1 to 59 had viability diminution at a 9.3 to 9.9 log10 CFU/g rate against the control at 11.30 log10 CFU/g. As an example, for experiment 40, with the proportion of 33.3IN:33.3L:33.3MX, a viability value of 9.85 log10 CFU/g was determined. Notwithstanding, we observed higher viability of *B. clausii* (9.8 log10 CFU/g) when we used L closer to the unit, as experiment 34 showed (0 IN:100 L:0 MX). Using a 50 L:50 MX ratio, *B. clausii* viability was 9.37 log10 CFU/g, corresponding to experiment 27 (0 IN:50 L:50 MX). In matrix I, when carrier agents were present in the same proportion and matched with experiment 32 (33.3 IN:33.3 L:33.3 MX), viability was 9.75 log10 CFU/g. Results showed that use of more than an encapsulating agent improved *B. clausi*i survival. It is worth mentioning that the standards presented in these experimental designs, particularly the three components of the Scheffe special cubic x special cubic model, surpassed the 6.0 log10 CFU/g minimum value recommended by the FAO/OMS (2002) to be considered probiotic, and correlated with previous reports from [31], who performed experiments with co-microencapsulate *Lactobacillus casei* and black-currant (Ribes nigrum)-extract anthocyanin dried via lyophilisation. These authors employed whey protein isolate (WPI), chitosan, and inulin at a 2:1:1 rate as carrier agents. They reported that viability of the powder was 11 log10 UFC/g as a starting value; after storage for 90 days at 4 °C, it reduced to 8.13-6.35 log10 UFC/g. Showing stability of carrier agent mixtures, Milea et al. [32] reported viability of co-microencapsulates via flavonoid lyophilisation, obtained from yellow onion peelings (Allium cepa) and *Lactobacillus casei* and employing whey protein isolate (WPI), inulin (I), and maltodextrin (MD) as carrier agents (2:1:1 proportion). These samples were encapsulated by lyophilisation at 1% and 2% concentrations probed into food one (cream cheese). The researchers reported results after storage of 21 days at 4 °C, recording a 6.6 and 7.41 log10 UFC/g viability at the concentrations mentioned above. Cayra et al. [33] suggested protection provided by L, IN, and MX materials for cellular structures of microorganisms via formation of crystals and water-molecule replacement in polar groups of cellular membrane lipids. For matrix II, we found a higher B. clausii viability value in two conditions: with use of a unit of G and in employment of a G and X mixture. This was seen, for example, in experiment 31 (0A:50G:50X composition) and experiment 3 (0 A:100 G:0 X). Obtained results showed that: I) Higher viability was obtained when IN and L and/or IN, L and MX at the same proportion were employed as carrier agents in matrix I, or, in matrix II, when G and/or G and X at the same ratio were used closer to the unit; II) Lower viability was observed with use of L and MX at the same rate in matrix I and with use of X closer to the unit in matrix II; III) The better combination of carrier agents and gums allowed better viability preservation; IV) We obtained one functional food, since the minimum viability value of 6.0 log10 CFU/g recommended by the FAO/OMS (2002) to be considered as probiotic was determined across the values in all experiments in this research.

### 3.3. Antioxidant Capacity Determination

Antioxidant capacity was determined with 2,2 diphenyl-1-picrilhidrazil (DPPH) radical inhibition using 5, 10 and 30 µg/mL (sample) as concentrations; as can be observed in Figure 3a–c, for matrix I, formed by IN, L, and MX, higher antioxidant activity (AA) was determined when the carrier agent was closer to the inulin unit and corresponded to experiment 33 (100 IN:0 L:0 MX), presenting concentrations of 56.06 AA to 5 µg/mL concentration, 58.75 to 10 µg/mL, and 84.35 to 30 µg/mL. Nevertheless, when the matrix was compounded by 50 IN:50 L, corresponding to experiment 15 (50 IN:50 L:0 MX), AA was 52.88 (5 µg/mL), 57.95 (10 µg/mL) and 81.51 (30 µg/mL), respectively. Samples that presented less AA were closer to the MX unit, as shown in Figure 3a–c, with activity of 38.58 at 5 µg/mL, 43.27 at 10 µg/mL, and 63.1 to 30µg/mL. These results correlate with observations reported by other authors, such as Martins et al. [34], who performed a lemongrass (Cymbopogon citratus (DC.) Stapf) essential oil micro-encapsulation study; the object of that work was to evaluate development, characterisation and production of particle antioxidant potential. Three different formulations were generated for essential oil encapsulates: M1 (5% of essential oil), M2 (10% of essential oil), and M3 (15% of essential oil). Each mixture used maltodextrin MD (DE 20) and gelatine (GEL) at a 4:1 (*w*/*w*) ratio as encapsulating agents. Result emulsions were lyophilised under 0.011 mbar and −60 °C conditions for 48 h. Concerning antioxidant activity, the authors reported that generally, due to a variety of presences of bioactive compounds, functional groups, and polarities as parts of the essential oil, the antioxidant effect had a starting antioxidant capacity value of 22.16 ± 0.04 mg TE/g, measured by the DPPH method. After lyophilisation, samples presented antioxidant potential of 2.46 ± 0.12 mg TE/g for M1, 7.74 ± 0.05 mg TE/g for M2, and 12.10 ± 0.30 mg TE/g for M3. Results showed that MD (DE 20) use influenced antioxidant capacity. Azarpazhooh et al. [35] evaluated pomegranate (Punica granatum L.) grinds extracted via the DPPH method for antioxidant capacity (RSA). The process consisted of use of maltodextrin (MDX) in three proportions—5, 10, and 15%—as a carrier agent, as well as calcium alginate at 0.1/(*w*/*w*) at a 1:5 proportion. Researchers reported lower inhibitory concentration (IC50), at 0.56 mg/mL of MDX for a 15% sample. Against the 0.86 mg/mL IC50 MDX observed in a 5% sample, results indicated the influence of MDX concentration on RSA increase. Notwithstanding, the obtained results could have been influenced by anthocyanins and polyphenols present in the sample.

For matrix II, consisting of G:X:A, we show the results in Figure 4a–c. At the concentrations of 5, 10, and 30µg/mL, more favourable results were obtained using G and X at the same proportions without presence of A; e.g., for experiment 49 (0 A:50 G:50 X). Notwithstanding, when A was closer to the unit, AA was lower; e.g., for experiment 20 (100 A:0 G:0 X), corresponding to AA of 48.36 (5 µg/mL), 52.06 (10 µg/mL), and 81.93(30 µg/mL), as shown in Table 1. Mansour et al. [36] micro-encapsulated anthocyanin (AC) extracts obtained from raspberries via lyophilisation (Rubus idaeus L.). Evaluation revealed three different anthocyanin concentrations (0.025%, 0.05%, and 0.075%), two encapsulating agents (soy protein isolate (SPI) and gum arabic (GA) at 5% concentration *w/v*), and SPI and GA at 2.5:2.5 % *w/v* concentration. Antioxidant capacities observed for these compounds at 0.025% were 25% for SPI, 45% for GA and 35% for the SPI+GA mixture.

Rezende et al. [37] elaborated on industrial waste and steelyard pulp (Malpighia emarginata DC) micro-encapsulates. They employed gum arabic (GA) and maltodextrin (MD) mixture in the same proportion (1:1; *w*/*w*) as carrier agents. These researchers used the DPPH method to contrast antioxidant activity for industrial waste and steelyard pulp through the dry aspersion and lyophilisation processes. Antioxidant activity after the sample encapsulation process was 129.16 μM TE/g for lyophilised samples and 155.24 μM TE/g for samples dried via aspersion. Obtained results allowed identification of (I) carrier agent combinations and gums that allowed the best antioxidant activity percentage; (II) higher antioxidant activity obtained when the carrier agent used in matrix I was inulin, or G and X at the same proportion for the 5 and 10 µg/mL concentrations in matrix II; and (III) lower antioxidant activity with use of only maltodextrin in matrix I and with G closer to the unit in the matrix II. This last evidence is in line with results reported by researchers.

## 4. Conclusions

Through experimental D-optimal designs, we prepared a functional ingredient formed by *B. clausii* and quercetin for two matrices: matrix I, formed by IN:L:MX; and matrix II, formed by A:G:X. With regard to yield, using the three compounds with the same proportion for both matrices, we obtained a higher value (88.08%) for experiment 32 (33.3 IN:33.3 L:33.3 MX) and (33.3 A:33.3 G:33.3 X). For matrix II, we observed higher viability (9.8 log10 CFU/gto) with use of L closer to the unit corresponding to experiment 34 (0 IN; 100 L:0 MX). We obtained higher AA when inulin was used closer to the unit corresponding to experiment 33 (100 IN:0 L:0 MX), presenting as 56.05 at a 5 µg/mL concentration, 8.75 at 10 µg/mL and 84.35 at 30 µg/mL. Matrix II presented higher viability values in two cases: with use of G closer to the unit and in employment of a G and X mixture, e.g., 9.9 log10 CFU/g in experiment 31 (0 A:50 G:50 X) and 9.8 log10 CFU/g in 3 experiment 3 (0 A:100 G:0 X). Higher AA was yielded with use of G and X in equal proportions, corresponding to experiment 49 (0 A:50 G:50 X), with AA of 54.85 at a concentration of 5 µg/mL, 55.44 at 10 µg/mL, and 67.09 at 30 µg/mL. Synergism between two matrices occurred with use of a higher I and G proportion, corresponding to experiment 46, which featured a yield of 86.5%; viability of 9.52 log10 CFU/g9.52 log10 UFC/g; and AA of 54.85 at a 5 µg/mL concentration, 55.44 at 10 µg/mL, and 67.09 at 30 µg/mL. 

## Figures and Tables

**Figure 1 polymers-14-05225-f001:**
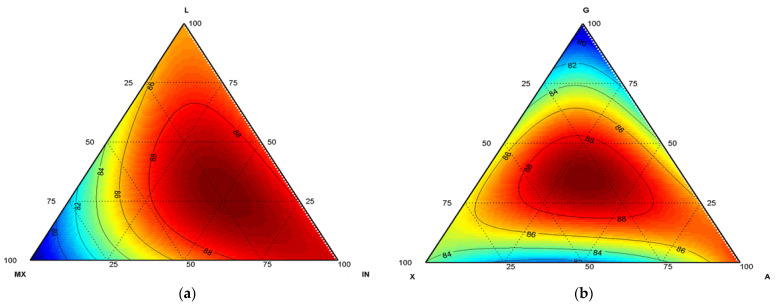
Yield surface graphic: (**a**) matrix I, IN:L:MX; (**b**) matrix II, A:G:X.

**Figure 2 polymers-14-05225-f002:**
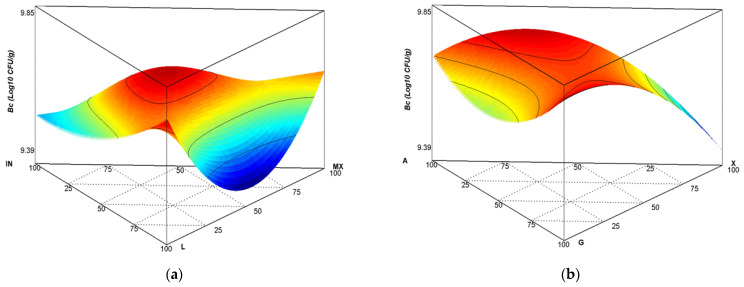
Graphics of surface responses to viability of *B. clausii* dried by lyophilisation, expressed as log10 (CFU/g). (**a**) Matrix I, IN:L:MX; (**b**) matrix II, A:G:X.

**Figure 3 polymers-14-05225-f003:**
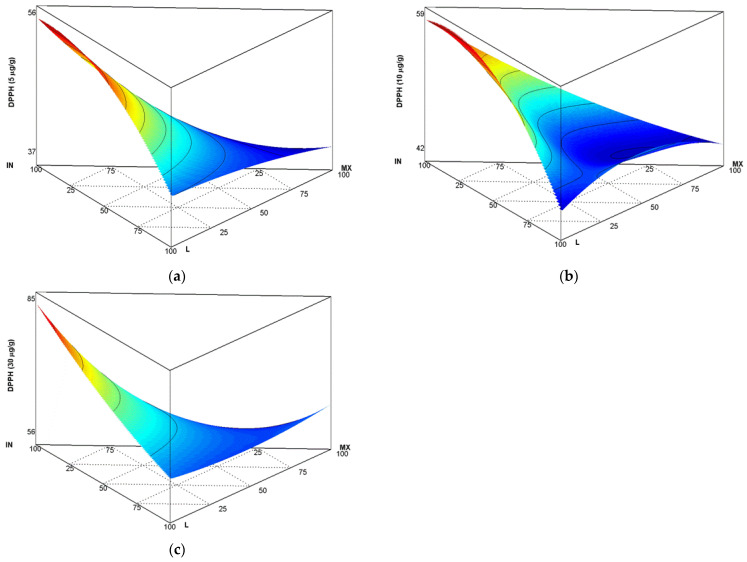
Antioxidant capacity with 2,2 difenil-1-picrilhidrazil (DPPH) radical inhibition for (**a**) 5 µg/mL, (**b**) 10 µg/mL, and (**c**) 30 µg/mL (sample), in matrix I (formed by IN, L, and MX).

**Figure 4 polymers-14-05225-f004:**
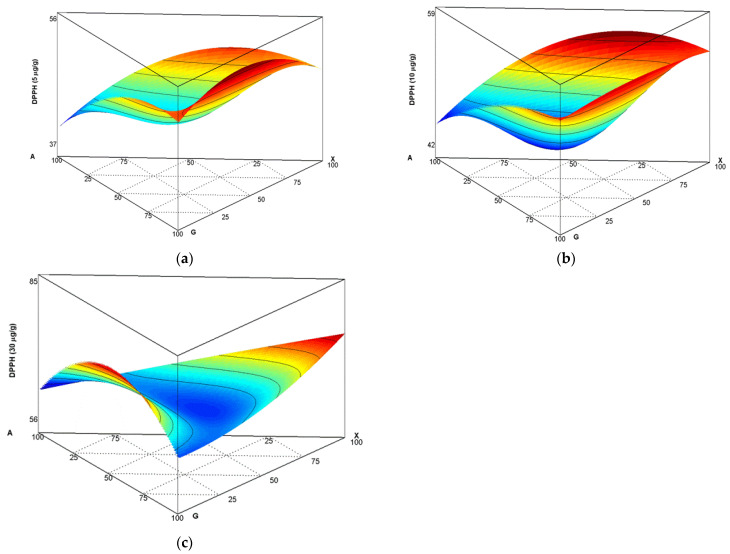
Antioxidant capacity of 2,2 difenil-1-picrilhidrazil (DPPH) radical inhibition: (**a**) 5 µg/mL, (**b**) 10 µg/mL, and (**c**) 30 µg/mL (sample), for matrix II, formed by A, G and X.

**Table 1 polymers-14-05225-t001:** Experimental design of two matrices for a special cubic x special cubic model.

		Matrix I			Matrix II			Yield	DPPH * Scavenging Activity			Bc
No	Run	IN	L	MX	A	G	X	(%)	5 µg/mL	10 µg/mL	30 µg/mL	(Log10 CFU/g)
1	44	100.0	0.0	0.0	0.0	50.0	50.0	82.7	52.35	55.7	89	9.7
2	15	0.0	100.0	0.0	100.0	0.0	0.0	82.6	52.32	61.7	76	9.67
3	51	0.0	100.0	0.0	0.0	100.0	0.0	87.6	52.45	56.6	80	9.6
4	3	0.0	100.0	0.0	0.0	0.0	100.0	86.1	51.91	55	72	9.52
5	20	0.0	100.0	0.0	50.0	50.0	0.0	87.8	56.05	58.8	84	9.48
6	57	0.0	100.0	0.0	50.0	0.0	50.0	86.6	51.45	55.4	71	9.56
7	48	0.0	100.0	0.0	0.0	50.0	50.0	84.4	48.36	52.1	82	9.3
8	32	0.0	0.0	100.0	100.0	0.0	0.0	83.9	53.74	55.5	79	9.3
9	35	0.0	0.0	100.0	0.0	100.0	0.0	87.6	52.6	56.6	71	9.3
10	41	0.0	0.0	100.0	0.0	0.0	100.0	89.2	52.3	55.3	70	9.48
11	11	0.0	0.0	100.0	50.0	50.0	0.0	86.4	49.86	60.2	77	9.43
12	18	0.0	0.0	100.0	50.0	0.0	50.0	86.7	50.18	53.1	71	9.52
13	39	0.0	0.0	100.0	0.0	50.0	50.0	87.2	54.85	55.4	67	9.48
14	25	50.0	50.0	0.0	100.0	0.0	0.0	84.6	48.55	51.2	75	9.64
15	43	50.0	50.0	0.0	0.0	100.0	0.0	86.3	51.8	58	78	9.6
16	47	50.0	50.0	0.0	0.0	0.0	100.0	88.1	41.95	42.9	63	9.75
17	8	50.0	50.0	0.0	50.0	50.0	0.0	93.5	50.06	55.1	72	9.73
18	40	50.0	50.0	0.0	50.0	0.0	50.0	79.9	49.69	51.2	77	9.37
19	27	50.0	50.0	0.0	0.0	50.0	50.0	87.9	48.36	52.1	82	9.87
20	19	50.0	0.0	50.0	100.0	0.0	0.0	82.2	51.63	59.6	68	9.82
21	1	50.0	0.0	50.0	0.0	100.0	0.0	76.5	51.22	52.8	63	9.56
22	23	50.0	0.0	50.0	0.0	0.0	100.0	86.8	49.2	51.2	71	9.75
23	7	50.0	0.0	50.0	50.0	50.0	0.0	74.6	50.68	53.3	63	9.37
24	34	50.0	0.0	50.0	50.0	0.0	50.0	77.2	38.58	43.3	63	9.67
25	55	50.0	0.0	50.0	0.0	50.0	50.0	89.1	50.68	58.5	76	9.6
26	45	0.0	50.0	50.0	100.0	0.0	0.0	82.9	49.41	54.9	75	9.37
27	37	0.0	50.0	50.0	0.0	100.0	0.0	81.1	49.32	54.3	76	9.52
28	2	0.0	50.0	50.0	0.0	0.0	100.0	85.7	42.19	49.8	59	9.56
29	30	0.0	50.0	50.0	50.0	50.0	0.0	84.5	39.42	45.4	61	9.43
30	54	0.0	50.0	50.0	50.0	0.0	50.0	85.6	51.21	53.6	81	9.52
31	58	0.0	50.0	50.0	0.0	50.0	50.0	88.3	38.58	44.1	62	9.7
32	16	33.3	33.3	33.3	33.3	33.3	33.3	85.8	48.18	57.9	74	9.56
33	5	100.0	0.0	0.0	33.3	33.3	33.3	81	51.13	56	68	9.75
34	56	0.0	100.0	0.0	33.3	33.3	33.3	81.8	53.27	54.5	72	9.6
35	24	0.0	0.0	100.0	33.3	33.3	33.3	79.8	51.82	58.7	78	9.48
36	6	50.0	50.0	0.0	33.3	33.3	33.3	86.5	54.85	55.4	67	9.52
37	28	50.0	0.0	50.0	33.3	33.3	33.3	81.2	50.78	57.3	79	9.37
38	29	0.0	50.0	50.0	33.3	33.3	33.3	78.9	50.7	54.5	65	9.85
39	31	33.3	33.3	33.3	100.0	0.0	0.0	82.5	51.76	58.3	71	9.67
40	38	33.3	33.3	33.3	0.0	100.0	0.0	86	52.05	57.3	83	9.48
41	26	33.3	33.3	33.3	0.0	0.0	100.0	84.2	48.45	54.3	78	9.7
42	14	33.3	33.3	33.3	50.0	50.0	0.0	84.1	54.85	55.4	67	9.73
43	33	33.3	33.3	33.3	50.0	0.0	50.0	83.4	52.88	58	82	9.43
44	42	33.3	33.3	33.3	0.0	50.0	50.0	86.4	49.05	54.3	72	9.73
45	52	100.0	0.0	0.0	100.0	0.0	0.0	82.8	53.06	56.9	80	9.67
46	36	100.0	0.0	0.0	0.0	100.0	0.0	81.2	55.24	57.6	74	9.48
47	53	100.0	0.0	0.0	0.0	0.0	100.0	84	52.88	58	82	9.67
48	59	100.0	0.0	0.0	50.0	50.0	0.0	85.2	45.12	54.7	79	9.78
49	13	100.0	0.0	0.0	50.0	0.0	50.0	81.2	53.87	57	72	9.7
50	22	66.7	16.7	16.7	66.7	16.7	16.7	84.5	54.36	56.8	69	9.64
51	12	66.7	16.7	16.7	16.7	66.7	16.7	83.4	40.24	52.4	70	9.88
52	17	66.7	16.7	16.7	16.7	16.7	66.7	86.2	51.91	52.8	63	9.67
53	4	16.7	66.7	16.7	66.7	16.7	16.7	81.1	52.01	52.7	70	9.73
54	50	16.7	66.7	16.7	16.7	66.7	16.7	85.7	43.39	43.9	60	9.6
55	21	0.0	0.0	100.0	0.0	0.0	100.0	88.5	53.18	55.4	65	9.37
56	10	0.0	0.0	100.0	100.0	0.0	0.0	86.4	40.26	42.3	62	9.8
57	46	0.0	0.0	100.0	0.0	100.0	0.0	87.7	46.75	53.9	75	9.6
58	9	0.0	0.0	100.0	50.0	0.0	50.0	85.2	40.38	42.6	62	9.9
59	49	0.0	0.0	100.0	0.0	50.0	50.0	86.8	53.38	56.8	71	9.56

* Free radical.

## Data Availability

Not acceptable.

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
