# Peer review of "Evaluation of Two Active System Encapsulant Matrices with Quercetin and Bacillus clausii for Functional Foods"

_polymers, 2022, doi:10.3390/polym14235225_

Round 1

Reviewer 1 Report (Previous Reviewer 1)

The authors did not adequately respond to the comments made in their article, so they must respond to each point in a timely manner. Also, the scientific quality of the manuscript is insufficient for publication in its current form. Specific questions and points requiring attention are itemized below.

1.      Reviewer comments: Line 53. Introduction section: “Antioxidants are added, obtaining different benefits, such as suppressing lipidic oxidation, increasing products' shelf life, and reducing the free radical concentrations inside of organism, consequently improving consumers' healthy”. Describe other examples of antioxidant molecules and their sources.

2.      Reviewer comments: Line 73. Describe other examples of probiotics microorganism and their sources.

3.      Reviewer comments: It is still difficult to find the novelty of the work concerning what has already been published. What is the difference between what is published with what the authors want to publish? It is not clear. The authors must describe these differences in the introduction section.

4.      Reviewer comments: Why do the authors describe in detail various wall materials?? All these wall materials will be used in the investigation???

5.      Reviewer comments: The introduction seems like a mini-review article and does not highlight the most critical information of the investigation. The authors must take us to the research problem they want to develop.

6.      Reviewer comments: The objective of the research is not clear. Please, revise the aim in detail.

7.      Reviewer comments: Line 176. What is the objective of passing through a sieve 1 mm the compounds of each matrix?

8.      Reviewer comments: Line 178. What Colony Forming Unit has 1 g of B. clausii???

9.      Reviewer comments: Line 190. Please, numbers in superscript

10.  Reviewer comments: Antioxidant activity: Why did the authors evaluate DPPH instead of ABTS? The authors should determine both assays.

11.  Reviewer comments: What standard was used for antioxidant activity? Calibration curve????

12.  Reviewer comments: Statistical analysis section must be added.

13.  Reviewer comments: How do the authors demonstrate the formation of microcapsules as an encapsulation of quercetin and B. clausii? The authors must display SEM images, displaying external and internal microstructural of the encapsulates.

14.  Reviewer comments: What optimal conditions to encapsulate quercetin and B. clausii were obtained?

15.  Reviewer comments: The authors did not display the statistical analysis of the experimental design. It is important to describe whether the model fits, and other information regarding the model.

Author Response

Please find attached responses to your comments 

without first thanking them for their review, which strengthened the research. 

Reviewer 2 Report (Previous Reviewer 3)

Latin name in lines 435, 437 & 440: Bacillus clausii ( c in clausii not in capital letter).

Overall is good.

Author Response

Encuentre las respuestas adjuntas a sus comentarios. 

Sin antes agradecerles por su revisión, que fortaleció la investigación. 

Round 2

Reviewer 1 Report (Previous Reviewer 1)

Comments must be attended

1.       Again, the introduction seems like a review article and does not highlight the most critical information of the investigation. The current state of the research field should be reviewed carefully, and key publications should be cited. Thus, the authors should rewrite the introduction section.

 2.       The manuscript did not show the SEM micrographs and discussion in the revised version.

Author Response

Thanks for the corrections

Round 3

Reviewer 1 Report (Previous Reviewer 1)

The article can be accepted

This manuscript is a resubmission of an earlier submission. The following is a list of the peer review reports and author responses from that submission.

Round 1

Reviewer 1 Report

The scientific quality of the manuscript is insufficient for publication in its current form. Specific questions and points requiring attention are itemized below.

 1.      Reviewer’s comment: The introduction was poorly written. It requires additional information on previous attempts when similar materials were used and what were the results.

2.      Reviewer’s comment: (line 73, page 2). Bacillus clausii in italics

3.      Reviewer’s comment: The objective of the research is not clear.

4.      Reviewer’s comment: It is difficult to understand the novelty of the work, also, the authors do not previously introduce us to the most important antecedents.

5.      Reviewer´s comment: What is the innovation of this paper? What is new in this work? It is not clear. The sentence should be added in the introduction section.

 6.      Reviewer´s comment: The authors must define the “microencapsulation process” as well as the wall materials more used to encapsulate bioactive compounds using lyophilization process.

 7.      Reviewer´s comment: Why do the authors used the freeze-drying process for encapsulation? Explain more in detail.

 8.      Reviewer´s comment: The section on lyophilizes preparation is not clear. Please, authors can rewrite the process in more detail to make it more understandable to the reader.

9.      Reviewer´s comment: The authors can do a process diagram of preparation of microcapsules??

 10.  Reviewer´s comment: How do the authors demonstrate that microorganisms were microencapsulated? The authors must add a result that demonstrate that microcapsules were obtained. For example SEM…

 11.  Reviewer´s comment: Statistical analysis section must be added.

 12.  Reviewer´s comment: Authors define that “the microencapsulation technique comprises the coating of small particles to form capsules with unique properties and different morphologies, which can reach diameters between nanometres to millimeters, protect bioactive ingredients against adverse reactions, and improve their functionality and bioavailability”. How do the authors demonstrate that microcapsules were obtained?

 13.  Reviewer´s comment: What is the particle size of the microcapsules obtained?

Reviewer 2 Report

The work is well written. the introduction and the figures and tables make reading pleasant. in my humble opinion the job can be accepted.

Reviewer 3 Report

There are some corrections:

1. It is notation IN, not I (line 19, 22, 24, 237, 318, 320)

2. Genrally the use of this citation is not mixed with the Vancouver model (line 62, 66, 76, 79, 186, 193, 221, 226, 232, 259, 271, 285, 286, 291)

3. Latin names written in italic (line 73, 105, 183, 184, 222, 227, 259, 272, 286, 291)

4. It is not clear. It's supposed to be one sentence, isn't it? ( line 285 - 289)

5. Figure 3 is supposed to be moved to line 279-280.

6. References writing is not consistent. Please pay attention to the Latin names (375, 377, 380, 400, 403, 412, 421)